# Curve Similarity Analysis for Reducing the Temperature Uncertainty of Optical Sensor for Oil-Tank Ground Settlement Monitoring

**DOI:** 10.3390/s23198287

**Published:** 2023-10-07

**Authors:** Tao Liu, Gang Liu, Tao Jiang, Hong Li, Changsen Sun

**Affiliations:** College of Optoelectronic Engineering and Instrumentation Science, Dalian University of Technology, Dalian 116024, China; liutao2020@mail.dlut.edu.cn (T.L.); newgun@mail.dlut.edu.cn (G.L.); jiangtao19980125@mail.dlut.edu.cn (T.J.); lihong@dlut.edu.cn (H.L.)

**Keywords:** optical fiber sensor, ground settlement monitoring of oil tank, temperature uncertainty, curve similarity analysis

## Abstract

A nonuniform temperature field can deteriorate the performance of sensors, especially those working in the field, such as an optical sensor for oil-tank ground settlement (GS) monitoring. In this case, the GS monitoring employs hydraulic-level-based sensors (HLBS), which are uniformly installed along with the oil-tank basement perimeter and are all connected by hydraulic tubes. Then, the cylinder structure of the oil tank itself can create a strong temperature difference between the sensors installed in the sunlit front and those in the shadow. Practically, this sunlight-dependent difference can be over 30 °C, by which the thermal expansion of the measuring liquid inside the connecting hydraulic tubes keeps on driving a movement and, thereby, leads to fluctuations in the final result of the oil-tank GS monitoring system. Now, this system can work well at night when the temperature difference becomes negligible. However, temperature uncertainty is generated in the GS sensors due to the large temperature difference between the sensors in the daytime. In this paper, we measured the temperature where the sensor was located. Then, we compared the results of the GS sensors with their corresponding temperatures and fitted them with two separate curves, respectively. After observing the similarity in the tendency of the two curves, we found that there was a qualitative correlative relationship between the change in temperature and the uncertainty in the sensor results. Then, a curve similarity analysis (CSA) principle based on the minimum mean square error (MMSE) criteria was employed to establish an algorithm, by which the temperature uncertainty in the GS sensors was reduced. A practical test proved that the standard deviation was improved by 73.4% by the algorithm. This work could be an example for reducing the temperature uncertainty from in-field sensors through the CSA method.

## 1. Introduction

Temperature effect is a major factor that must be considered in sensor development, especially for field application purposes. The temperature uncertainty can deteriorate the sensor performance and finally determine its highest potential, by which the accuracy of the sensors is intrinsically distinct from each other [1,2]. In a case of nonuniform temperature situation, this uncertainty becomes a prominent factor, such as GS monitoring of an in-service oil tank [3].

An oil tank can itself create a large temperature difference between the sunlit front regions and its shadows. According to practical measurements, this temperature difference can be over 30 °C in the daytime of summer. This means the HLBS can only accurately work at night, when all the sensors as well as the connected vessels are in an approximately homogeneous temperature field [4,5,6]. In order to overcome this problem, some techniques were developed to eliminate the daytime temperature uncertainty and could be summarized as two aspects roughly: hardware and software.

The hardware approach involved a slow circulation of the measurement liquid to facilitate the attainment of thermal equilibrium in the hydraulic tube [7]. However, the power supply has to meet the requirements of the fire-proof certification of oil-tank management. Another most prominent achievement was from the structural design of the distributed hydraulic tubes [8]. This design involved one main measuring hydraulic tube being accompanied by another compensation tube which is sensitive to all effects affecting the measurements. The compensation tube serves to isolate any interferences that may arise. Both measuring and compensation tubes were filled with the same measuring liquid and configured to be surrounded by six auxiliary tubes. In addition, all the tubes were well covered and assembled as one functional hydraulic tube. This design can surely well compensate for the temperature effect caused by the hydraulic tube. However, an assembly tube containing six liquid-filled sub-tubes makes connecting tough and the installment of the sensor cumbersome. Thus, until now, there has been no delicate hardware innovation being developed to replace the simple conventional HLBS in the practical oil-tank GS monitoring system.

The software algorithms to overcome the temperature uncertainty were commonly from theoretical calculation [9], simulation [10], neural networks [11,12], etc., to relate the GS results to a temperature model. Wu. et al. studied both the measuring liquids and hydraulic tube materials in laboratory condition and then transferred these to field applications [13]. This was suitable for situations where the sensors are all nearly equal with small temperature variations; it is difficult to use in the case of a large temperature difference. 

The most promising method to deal with the nonuniform temperature cases was CSA, which was originally developed for pattern recognition and now has become popular in many fields [14,15,16]. Brink et al. employed the CSA method to figure out how the ecosystems evolve with environmental factors [14]. This highlighted the potential of CSA in addressing temperature uncertainty and inspired this work directly. 

In this paper, we employed the CSA method to reduce the temperature uncertainty of optical GS sensor for oil-tank monitoring. The two curves were generated using the data obtained from the practical oil-tank GS sensors, which were configured in HLBS, and the measured temperatures at each sensor location. An algorithm stemming from the CSA theory was developed to reduce temperature uncertainty from the results of the GS monitoring.

## 2. Temperature Uncertainty of In-Field Oil-Tank GS Sensor

A practical problem of temperature uncertainty existed in oil-tank GS monitoring due to sunlight irradiation on a large cylindrical structure of the oil tank. In this case, an oil tank with 100,000 m^3^ in its capacity bears a diameter of 80 m and a height up to 21.8 m. A large temperature difference can be produced between the regions in the sunlit front and those in the shadow, as indicated in Figure 1a. In summer, this difference can exceed 30 °C, which is measured by the distributed Raman temperature sensor (HG-DTS-160, from Tianjin Huigan Optoelectronic Technology Co. Ltd., Tianjin, China. The range is −40 °C~+100 °C, with a temperature accuracy of ±1 °C and a resolution of 0.1 °C. It is the temperature difference that causes the uncertainty in the results of oil-tank GS. In optical temperature measurement, many methods have been proposed, such as liquid-filled photonic crystal fiber [17], blackbody radiation sensors [18], fluorescence-based sensors [19], etc. [20]. However, the most commercial and wide-used in oil tank monitoring is the distributed Raman temperature sensor, which can be found in market. 

The nonuniform distribution of temperature around the oil tank keeps on driving a thermal movement in both the GS sensor container and the measuring liquid inside the hydraulic tubes. The HLBS was composed by nine GS sensors, which were hydraulically connected by a steel tube with an inner diameter of 19 mm. The installment is described in Ref. [5]. The eight GS sensors of these were installed uniformly around the perimeter of the oil-tank base, while the other one acted as a reference sensor, which was fixed on a benchmark, as shown in Figure 1b. 

Eight sensors can acquire the GS information of the oil tank clearly, which is vital for further analysis. The installation position is premeasured by a total station to ensure the installation position being on the same level. The HLBS can give a common reference for all sensors installed around the tank. In addition, the optical part based on low coherent interferometry is taking in charge of high-precision measurement. The temperature effect can be well avoided through a well-controlled low coherent measurement, which is only sensitive between the projecting lens and the measuring liquid surface, where a weak reflection of light can be examined. The optical interrogation system was designed to survey each GS sensor from the control center in a distance of about one kilometer through a cable of single-mode optical fiber. For further detailed information on the principle of the low coherent interferometry used in this system, readers can refer to the publications Refs. [5,21,22]. 

The demodulation methods for hydrostatic leveling GS techniques can be found as the ultrasonic method, string vibration method, fiber Bragg grating (FBG) sensor, low coherent interferometry, etc. However, the first two methods require electrical power, which can be a significant barrier for their usage in the circumstance of oil tanks. In addition, FBG-based sensors are inherently temperature-sensitive, which limited their practical application in an environment with a large temperature difference. The low coherent interferometry bears advantages such as no electrical risks, a wide measurement range, high accuracy, and the sensor itself being insensitive to temperature changes.

The average power consumption of this system is about 35 W, but it is worth noting that all the electrical components are installed in the control center, ensuring that power consumption does not pose a problem for the sensors. The delay time, on the other hand, of interrogation for each sensor is about 10 s, about 3 min for nine sensors. It is slower than the other kinds of sensors, but that is enough for the GS measurement which is often calculated by year.

The sensor probes around the oil tank are deliberately designed without any electricity. This can eliminate potential electrical hazards and ensures a safer operation in the oil tank. The delay time for each channel of the sensor is approximately 10 s, resulting in a total delay of just over a minute for all nine channels. This delay is not a significant issue, particularly for large oil tanks, as it does not significantly impact the overall measurement process. The readouts of the GS sensors are given in Figure 1c, in which there is a temperature uncertainty characterized as a large fluctuation in GS measurement results, which could be obviously related to sunlight irradiation. In Ref. [5], we took the midnight results as a measure of the oil-tank GS due to the slow variation property of oil-tank GS, by which the GS information was analyzed in accompany with an oil-in procedure. In this paper, we aimed to reduce this temperature uncertainty by using a technique of CSA in order to make the HLBS work well in daytime.

## 3. The CSA for Reducing the Temperature Uncertainty of Oil-Tank GS Monitoring

In this section, we construct a CSA-based algorithm to eliminate temperature uncertainty existing in the daytime GS readouts, as given in Figure 1c. Two curves were plotted as the GS of an oil tank and the temperature at which the sensor was located.

### 3.1. The Two Curves

Two curves of GS measurement and temperature changes were plotted in one figure for comparison. The GS results were recorded by GS sensors and the temperatures by the distributed Raman fiber temperature sensor. The GS information and the temperatures were simultaneously interrogated and were presented in Figure 2. 

In Figure 2, we can observe the similar tendency of the nine GS sensors and their corresponding environmental temperature variations, even if the phase of the temperature changes does not exactly match the fluctuations in the GS. 

From Figure 2a–i, we find that the two curves from different sensors show different similarities, which could be ascribed to the location causes. The time of peak appears at around 14:00, except sensor #7 of (g), which is located at the east side of the oil tank and it is the first sensor to be irradiated by the sunrise in the morning. Sensor #8 of (h) shows a delayed time at 16:00 in the blue curve and this corresponds to a glancing incidence of sunset. These results prove that the uncertainty of GS results, as given by the blue curves in Figure 2, can be related to these red curve temperatures. Then, we are going to set up a quantitative relationship between the two curves based on the principle of the CSA to reduce this temperature uncertainty.

### 3.2. The CSA

The CSA underlying the criteria of MMSE was employed to build up a description to quantitatively relate the temperature uncertainty of GS to the temperature changes by minimizing their variances before establishing a mapping model. The CSA is a technique widely used in pattern recognition and other fields [23,24,25,26]. Its essential role is to describe the relationship of the two curves in a mathematical model. 

The temperature changes can be defined by a dataset $\Delta T_{j}$ and GS by $\Delta H_{j}$ for each sensor, written as below:(1)$$\Delta T_{j}={\left[\delta T_{j,1},\delta T_{j,2},\ldots ,\delta T_{j,n}\right]}^{T}\in R^{n}$$
(2)$$\Delta H_{j}={\left[\delta H_{j,1},\delta H_{j,2},\ldots ,\delta H_{j,n}\right]}^{T}\in R^{n}$$
where j is the index of GS sensor and n is the number of data involved in calculation. Then, we seek an estimated function to satisfy the equation $\Delta H_{j}=f(\Delta T_{j},W_{j,z})$, where $W_{j,z}=\{k_{j}^{\left(z\right)},k_{j}^{\left(z-1\right)},\ldots k_{j}^{\left(0\right)}\}$ contains coefficients of sensor j, and z is the order of polynomial. $\Delta H_{j}$ is a dataset composed by the readouts of all the GS sensors given by the blue curves in Figure 2. Each element is listed as $\delta H_{j,n}$, where j is the index of GS sensor and n is the number of data involved in calculation. The MMSE estimation can be expressed by using a minimization operator (arg min) as below:(3)$$E{\left({\hat{W}}_{j,z}\right)}_{MMSE}=arg\;\underset{W_{j}}{\min}L(\Delta H_{j},f(\Delta T_{j},W_{j,z}))$$
where E is the estimation function and L is a loss function, which can be applied to figure out the function $f(\Delta T_{j},W_{j,z})$. The experimental results in Figure 2 showed the correlative relationship between the two curves. As Figure 3 shows, a first-order polynomial fitting can well fit the relationship of $\Delta H_{j}$ with $\Delta T_{j}$. When the temperature difference tends to zero, the temperature uncertainty of GS should also become zero. In this case, the intercept of the first-order polynomial is equivalent to zero. As a result, the dataset $W_{j,z}$ only includes one first-order coefficient $k_{j}$ for sensor j. Thus, the loss function L can be simplified as:(4)$$L(\Delta H_{j},f(\Delta T_{j},k_{j}))=\sum_{i=0}^{n}{\left[\delta H_{j,i}-f\left(\delta T_{j,i},k_{j}\right)\right]}^{2}$$

The task of the designed algorithm is how to optimize the coefficient $k_{j}$ to minimize the loss function L and, at that time, the function $f(\Delta T_{j},W_{j})$ becomes an optimal linear fitting as well. Thus, a relationship of $\Delta T_{j}$ with $\Delta H_{j}$ for each sensor can be expected as:(5)$$\Delta H_{j}=k_{j}\Delta T_{j}$$
where $k_{j}$ is the slope of fitting function and it can be used to describe how the environmental temperature changes can induce a temperature uncertainty in GS sensor. As an example, we presented four of the nine sensors by choosing each from the different directions of the sensor locations around the oil tank. The results are presented in Figure 3. 

In Figure 3, we can find that the similar tendency can be quantitatively described by the proposed model. We extracted the $k_{j}$ illustrated in Equation (5) and R-square of the data ($R_{j}^{2}$) of sensor *j* = 1, 2, 3…, 9 from calculation, respectively, and these are listed in Table 1, in which $R_{j}^{2}$ is defined as $R_{j}^{2}=1-\frac{\sum_{i=1}^{n}{\left(\delta H_{j,i}-k_{j}\delta T_{j,i}\right)}^{2}}{{\sum_{i=1}^{n}\left(\delta H_{j,i}-\bar{\delta H_{j,i}}\right)}^{2}}$, which is used to evaluate the linear correlative relationship and $\bar{\delta H_{j,i}}$ is the average of $\delta H_{j,i}$ of sensor j. If the perfect correlative relationship is satisfied, $R_{j}^{2}$ must equal 1. 

In Table 1, the minimum of $R_{j}^{2}$ happened at sensor #2, that is 0.514. This indicates the existence of a weak positive correlation between $\Delta H_{j}$ and $\Delta T_{j}$ in terms of the overall trend. Thus, Equation (5) is experimentally proved and, in the following section, we are going to analyze the underlying reason and design an algorithm to reduce this temperature uncertainty from the results of $\Delta H_{j}$.

### 3.3. The Correlation Origin of $\Delta H_{j}$ with $\Delta T_{j}$

In discussion of the origin of the correlative relationship of $\Delta H_{j}$ with $\Delta T_{j}$ of sensor j, the actual configuration can be simplified to a model of HLBS configured in two simple tanks connected by a tube. If one of the tanks is heated and the other is kept at a constant temperature, the liquid level inside the tank must rise due to the increased temperature inducing a volume expansion; in the meantime, the liquid density must be correspondingly decreased. In this case, the liquid pressure P at the bottom of the heated tank should still obey the basic physical law: $P=\rho gh=\frac{\rho shg}{s}=\frac{\rho Vg}{s}=\frac{mg}{s}$, where ρ is the density of liquid, h is the liquid height, s is the cross-section area of the tank, m is the mass of liquid in the tank, and V is the volume of liquid in the tank. Thus, the liquid pressure P will not be changed with the increases in liquid height, h, due to the decreases in density, ρ, being counterbalanced in a large part. This leads to the isolation of the liquid within the sensors and prevents it from flowing out of the tube connected at the bottom. So, the changes in h are not easily balanced immediately, and this is the reason why temperature uncertainty existed in the result. In the following section, an algorithm is designed to reduce this uncertainty based on the CSA.

### 3.4. The Algorithm for Reduction in Temperature Uncertainty According to the CSA

In order to develop an effective algorithm to deal with the temperature uncertainty which deteriorated the GS performance, we combined the correlative relationship given in Equation (5) with the intrinsic slow property of the GS of the oil tank. Figure 5 in Ref. [5] demonstrated that the GS variation was about 7 mm in the time period of 24 h during a whole oil-in procedure. As to the effect of temperature, the practical recording given in Figure 2d shows that the temperature uncertainty can reach over 60 mm! So, extracting the real GS information must well deal with the temperature effect, except the GS monitoring was carried out at night only.

On the other hand, coefficients $k_{j}$ in Table 1 indicate that the different sensors at different locations, as shown in Figure 1b, can exhibit significant diversity. This could be from the GS sensor installation, the orientation of the specific sensor, etc. Anyway, these coefficients provide a weight for each sensor in the algorithm to compensate the temperature uncertainty one by one, which forms the specialty to separate one sensor from another.

After continuously collecting the GS information and temperature data, we need to add them to a fixed-length sequence; this algorithmic structure enables the calculation of up-to-date parameters. The property of this structure has a moving average window, such as Kalman filter or the cyclic convolution algorithm. The flowchart of the algorithm is illustrated in Figure 4, which implemented two functions: the CSA and uncertainty reduction. 

Firstly, the coefficients $k_{j}$ are calculated according to Equation (5) based on a continuous 24 h practical acquisition of GS and temperature simultaneously. All the data were acquired in a one-hour interval, so 24 data were collected from each sensor per day. A CSA algorithm is developed based on these recorded data, and the predicted temperature uncertainty can be figured out as the coefficient multiplied by the temperature changes, and that is $k_{j}\delta T_{j}$. Thus, the temperature uncertainty can be subtracted from the practical GS results, $\delta H_{j}$, underlying the relationship given by Equation (6) as:(6)$$\tilde{H_{j}}=\delta H_{j}-k_{j}\delta T_{j}$$
where $\delta H_{j}$ is a readout of GS sensor j, $\delta T_{j}$ is the temperature changes measured at the location of sensor j, and $\tilde{H_{j}}$ is the expected GS results after reducing the temperature uncertainty. 

As time goes on, new data are recruited once an hour and the length of the data sequence in 24 h keeps constant, and the data will be operated according to the principle of first-in first-out, as shown in Figure 5.

In Figure 5, the previous 24 h GS sequence is 0.1, 0.1, 0, …, 0.4, 0.5, 0.3, and the temperature changes are 3, 2.4, 1.7, …, 5.8, 5.7, 3.9, correspondingly. The data sequence was organized in accordance with the stack structure. Suppose a new datum is recruited at the right end; it will push all the 24 data to be moved to the left and the leftmost datum will be out of the stack, as shown in Figure 5. This forms the present data sequence as 0.1, 0, −0.1, …, 0.5, 0.3, 0.1 for GS sequence and 2.4, 1.7, 1.5, …, 5.7, 3.9, 3.5 for temperature changes, respectively. If this process continues, a new $k_{1}$ based on the new data sequence can be calculated according to Equation (5) as 0.076. After, instead of inserting this result into Equation (6), the revised result represented by $\tilde{H_{j}}$ can be calculated as $\tilde{H_{j}}=0.1-0.076\times 3.5=0.084$. The performance of this algorithm will be tested by embedding the algorithm into a practical GS sensor in the following section.

## 4. The Practical Test

### 4.1. Data Comparison

To test the performance of the designed algorithm, a practical oil-tank GS monitoring was carried out. In order to prove the effectiveness of the proposed algorithm, a result shows how the GS sensor can work in a nonuniform temperature case, such as daytime, just like that in uniform at midnight. Due to the temperature uncertainty of GS being mainly characterized by the nonuniform temperature distribution in space, a comparison result is used to show how the GS sensor works with or without the algorithm was necessary. We extracted the GS information at a specific time 16:00, as marked by the blue arrow in Figure 1c and deployed in space according to the defined sensor index number in Figure 1b. The environmental temperature at each sensor was recorded with their spatial locations, as given in Figure 1b as well, and the process was carried out as described above. The results of the GS sensor work with or without the algorithm are demonstrated in Figure 6. 

Both the GS results and the environmental temperature were recorded and plotted corresponding to their spatial locations, as given in Figure 1b. The GS monitoring from each sensor was processed by the algorithm to reduce the temperature uncertainty in that specific moment. In addition, the results are shown in Figure 6.

In Figure 6, from the raw GS measurement, it can be observed that the significant fluctuation in GS can reach above 50.0 mm! That can be attributed to the nonuniform temperature field for different sensor locations between the sunlit front and shadow described above. Then, the proposed algorithm can reduce the temperature uncertainty and make the GS sensor work well under such a large temperature difference. It can be seen that sensors #3, #4, #5, and #8 are in the sunlit front area, sensors #1, #2, and #9 are in the shadow, and sensors #6 and #7 are in the shadow of a neighboring tank, considering the location illustrated in Figure 1b. When the temperature rises, the liquid expands. In addition, it makes the liquid level rise, therefore, leading to a large readout fluctuation. The larger temperature fluctuation, the larger the fluctuation in GS measurement. That is the reason for a large fluctuation in the measured data. 

As for the algorithm, it builds a relationship between the temperature uncertainty (fluctuation in GS) and temperature changes. When the relationship is built and temperature is measured, the temperature uncertainty can be predicted and it is filtered in a large part. That is the reason for the large difference in the sun-shined sensors between the algorithm and the measured data in Figure 6. Then, the proposed algorithm can reduce the temperature uncertainty and make the GS sensor work well under such a large temperature difference. 

Further analysis showed that the standard deviation was able to be reduced from 18.8 mm to 5.0 mm, so the improvement in terms of deviation is about 73.4%. On the other hand, the difference between the maximum and the minimum among the results decreased from 51.1 mm to 14.2 mm, and this improvement is about 72.2%. These results proved the effectiveness of the designed algorithm.

### 4.2. Further Improvement

The proposed algorithm somehow improved the performance of GS sensor. Further improvements can consider the following aspects:

Increase the accuracy of the temperature measurement. The Raman temperature measurement system can measure the distributed temperature along with the perimeters of the oil-tank base, but its accuracy is quite limited due to the temperature-sensing fiber not being embedded in the measured liquid. 

On the other hand, the theory is established based on the assumption that the sensors of HLBS are accompanied by the temperature right at the sensor location. In practice, the temperature is a distributed variable and it is a function of the spatial locations along with the oil-tank perimeter. So, more complex temperature considerations are needed to approach the high-accuracy goal. 

## 5. Conclusions

In this paper, the temperature uncertainty of the GS sensor of an oil tank was studied by using the CSA method. Two curves were constructed based on the readouts of the GS sensor and its corresponding environmental temperature changes. The constructed algorithm was developed based on the CSA principle. Finally, a practical test was carried out to prove the effectiveness of the proposed algorithm. This work could become an example for the temperature performance improvement on in-field sensor development. 

## Figures and Tables

**Figure 1 sensors-23-08287-f001:**
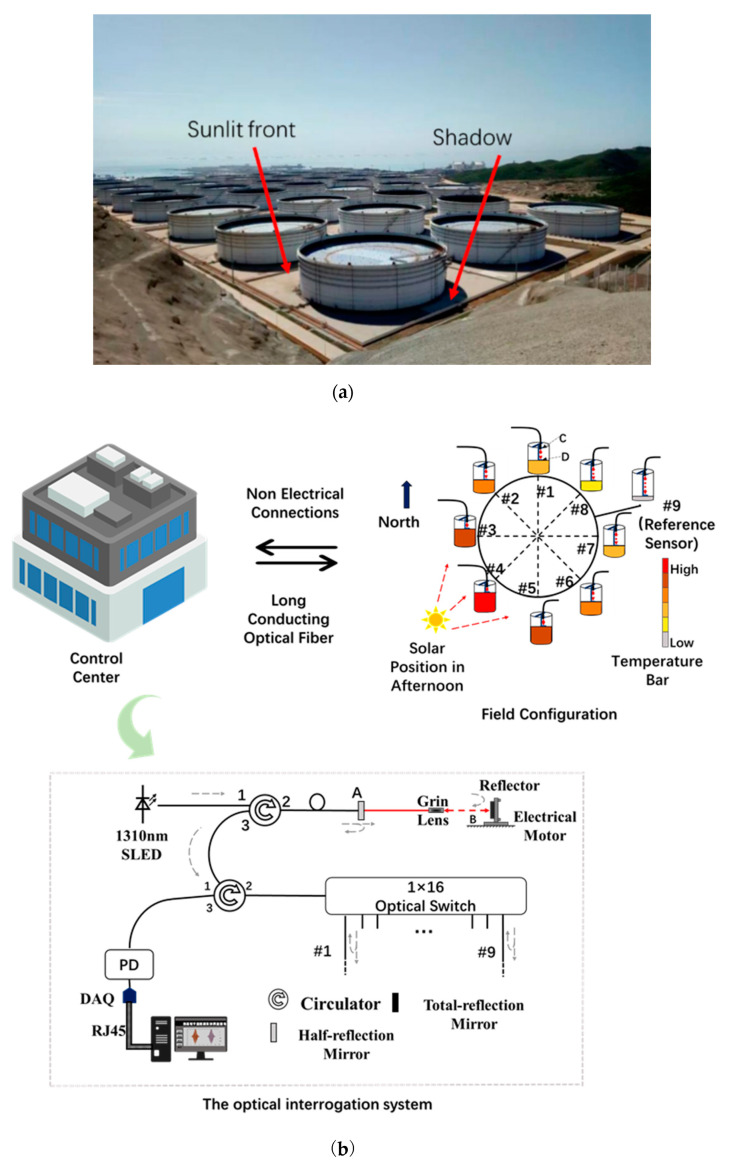
Temperature uncertainty in sensor readouts between the day time and night. (**a**) Indication on the regions of sunlit front and shadow formed by the cylindrical oil tank; (**b**) the nonuniform temperature distributed around the oil tank base, the location of the control center and the optical interrogation system; (**c**) the readouts of each sensor vs. time; the temperature uncertainty was observed and could be directly related to sunlight irradiation. The blue arrow indicates a moment with large temperature uncertainty of all sensors and it will be analyzed in the following section.

**Figure 2 sensors-23-08287-f002:**
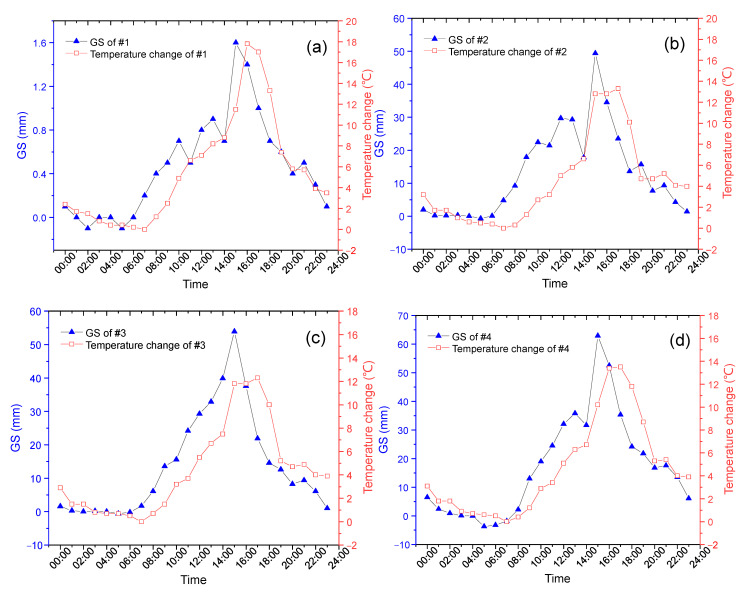
The fluctuations in oil-tank GS results compared with their environmental temperature changes. (**a**–**i**) represent the results of sensor #1 to sensor #9, respectively.

**Figure 3 sensors-23-08287-f003:**
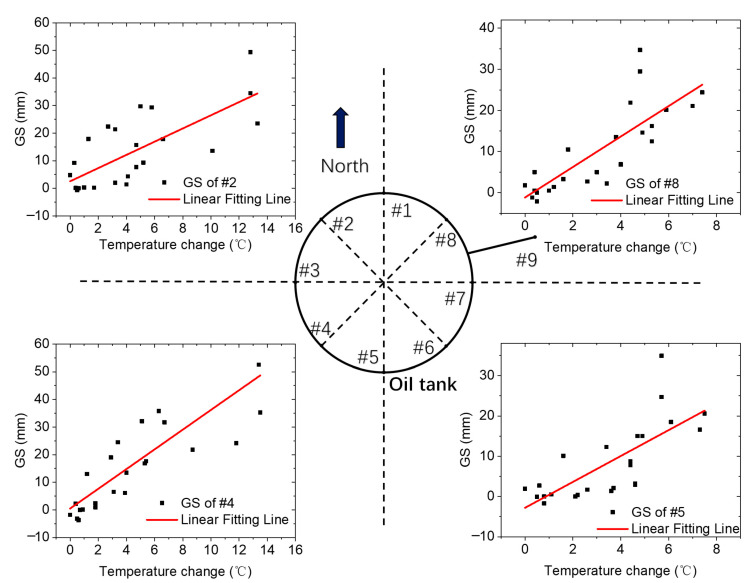
A correlative relationship of $\Delta H_{j}$ with $\Delta T_{j}$. Four of the nine sensors were selected to illustrate how the differences between the two the curves distributed around the oil tank.

**Figure 4 sensors-23-08287-f004:**
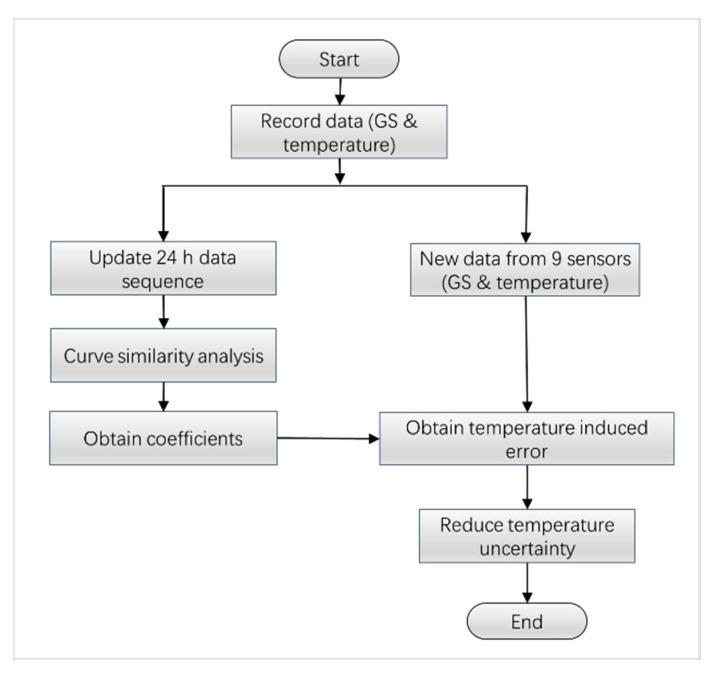
The flow chart of the proposed algorithm for reducing the temperature uncertainty.

**Figure 5 sensors-23-08287-f005:**
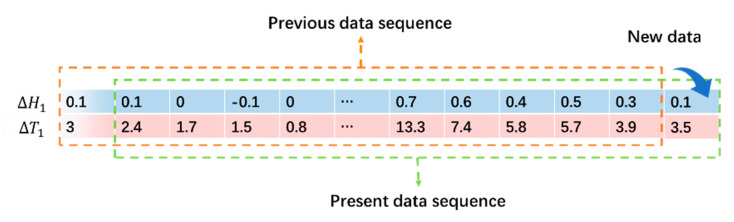
An example to illustrate how the proposed algorithm is operated according to the first-in first-out principles.

**Figure 6 sensors-23-08287-f006:**
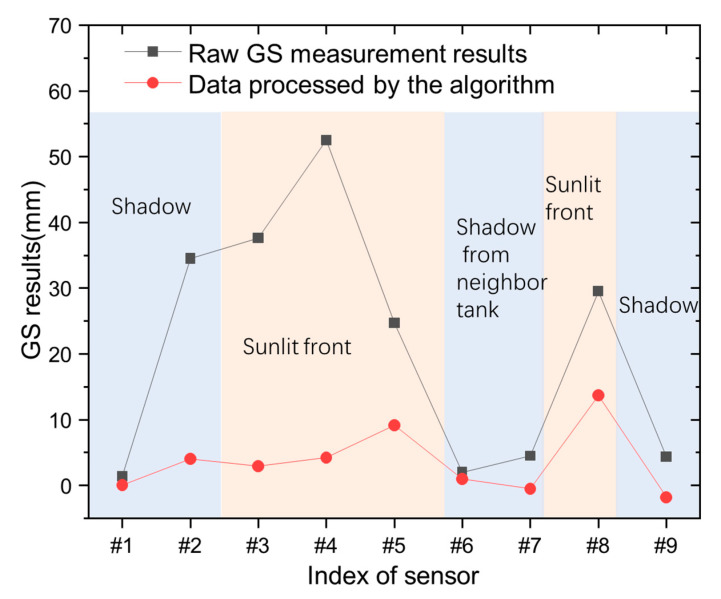
The results from sensors consisting of the raw GS measurement results and data processed by the algorithm in different sunshine conditions.

**Table 1 sensors-23-08287-t001:** The calculated coefficient $k_{j}$ according to the expected Equation (5), with their $R_{j}^{2}$.

Sensor	$k_{j}$	$R_{j}^{2}$
#1	0.076	0.724
#2	2.395	0.514
#3	3.147	0.601
#4	3.571	0.681
#5	3.218	0.534
#6	0.331	0.561
#7	1.407	0.566
#8	3.706	0.654
#9	0.581	0.717

## Data Availability

The data generated or analyzed as part of the research are not publicly available. This research keeps going on and the data will be disclosed with the permission of the oil-tank-running corporation.

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
