# Peer review of "Curve Similarity Analysis for Reducing the Temperature Uncertainty of Optical Sensor for Oil-Tank Ground Settlement Monitoring"

_sensors, 2023, doi:10.3390/s23198287_

Round 1
Reviewer 1 Report
In this paper, the authors have considered a very practical aspect of temperature uncertainty that is introduced due to the ambient temperature that the optical sensor is exposed to. The ensuing nonuniformity in the temperature field can deteriorate the performance of the sensors, especially those working in the field, such as an optical sensor for oil-tank ground settlement (GS) monitoring.
In order to address this issue, the authors have very cleverly adapted the already tested algorithm of Curve Similarity Analysis (CSA), and have presented their findings, which are certainly of statistical significance and prove their point. Moreover, the authors have also considered the practical case of considering the ground-based sensor data so as to showcase the applicability of their method to address the temperature non-uniformity and its effects on the oil tank ground settlement monitoring on field.
The authors have presented their research and supporting data well and have also explained their algorithm is an appreciable manner. Technically, I am quite satisfied with the treatment of this practical problem and also the supporting research to address this issue and hence I recommend publication of this paper in sensors.
I would recommend the authors to have this professionally proofread so as to improve the English sentence formation.
Reviewer 2 Report
Dear Editor,
In this paper, a setup with 9 sensors was designed for certainty in measurement. The topics of the manuscript is interesting and the results support the title and the abstract. The manuscript needs to be revised to reach the acceptance level So the following comments should be addressed:
1. Could you specify the temperature measurement range of the setup?
2. Was there an examination of the impact of humidity and pollution on the system's performance?
3. It appears that the system can produce acceptable results with fewer sensors. What criteria were used to determine the selection of 9 sensors?
4. The rationale behind designing the proposed system needs further clarification. It should be discussed in the introduction section.
5. The manuscript lacks an explanation for choosing the specific optical sensors used in this work.
6. In the introduction section, it would greatly benefit the readers to present a scenario for introducing optical sensors. Please consider discussing optical sensors based on junction, plasmonic, and photonic crystal technologies, and comparing their features to provide a rationale for selecting the current sensor. The following papers help in enhancing the introduction: Optik 168 (2018) pp. 342-347, Sensors 22(15) (2022) 5722, Diamond and Related Materials 131 (2023) 109594
7. Power consumption and delay time are important parameters that should be provided and compared with other systems.
Kind regards,
Reviewer 3 Report
The article is well organized and written, and is algorithmically innovative. However, there are still some minor issues, as follows:
1.In all Figure, the temperature unit is not displayed properly.
2.How to explain the large difference in the light-facing surface position between the algorithm and the measured data in Figure 6?
The article is well organized and written.
Round 2
Reviewer 2 Report
ِDear Editor,
The comments have been addressed so I can recommend the manuscript for publication.
with kind regards,